# The Curious Case of Hallucinatory (Un)answerability: Finding Truths in the Hidden States of Over-Confident Large Language Models

**Aviv Slobodkin[1], Omer Goldman[1], Avi Caciularu[1,2], Ido Dagan[1], Shauli Ravfogel[1]**

[1]Bar-Ilan University    [2]Google Research
{lovodkin93, omer.goldman, shauli321}@gmail.com
avica@google.com
dagan@cs.biu.ac.il

## Abstract

Large language models (LLMs) have been shown to possess impressive capabilities, while also raising crucial concerns about the faithfulness of their responses. A primary issue arising in this context is the management of (un)answerable queries by LLMs, which often results in hallucinatory behavior due to overconfidence. In this paper, we explore the behavior of LLMs when presented with (un)answerable queries. We ask: do models *represent* the fact that the question is (un)answerable when generating a hallucinatory answer? Our results show strong indications that such models encode the answerability of an input query, with the representation of the first decoded token often being a strong indicator. These findings shed new light on the spatial organization within the latent representations of LLMs, unveiling previously unexplored facets of these models. Moreover, they pave the way for the development of improved decoding techniques with better adherence to factual generation, particularly in scenarios where query (un)answerability is a concern.[1]

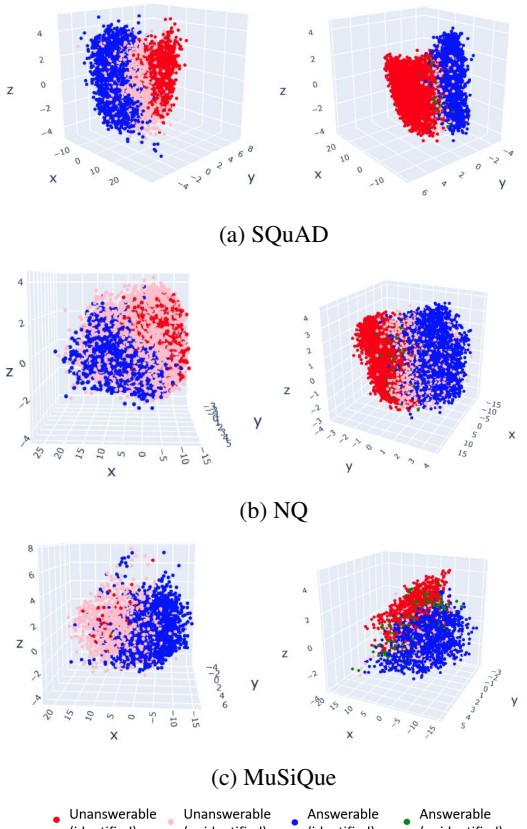

(a) SQuAD

(b) NQ

(c) MuSiQue

- Unanswerable (identified)
- Unanswerable (unidentified)
- Answerable (identified)
- Answerable (unidentified)

Figure 1: 3D PCA projection of the last hidden layer's embedding of Flan-UL2 on each of the three benchmarks. The left images show the embeddings with the regular prompt, and the right ones — with a hint-including prompt. Blue and red dots are examples correctly detected by the model as answerable and (un)answerable, respectively, while the pink dots are for (un)answerable examples that the model provided answers to. The figures show the good separability between the three groups.

## 1 Introduction

Modern large language models (LLMs) have been tantalizing the NLP community in the last couple of years (Brown et al., 2020; Chen et al., 2021; Chung et al., 2022), demonstrating great potential for both research and commercial use, but these models are of course not problem-free. Among their unfavorable behaviors it is possible to find toxicity (Welbl et al., 2021; Deshpande et al., 2023), bias (Nadeem et al., 2021; Abid et al., 2021), and hallucination (Manakul et al., 2023; Ji et al., 2023).

One of the settings in which LLMs are notoriously prone to hallucinate is when presented with (un)answerable questions (Sulem et al., 2021; Asai

and Choi, 2021; Amayuelas et al., 2023). Recent works in this setting, which is the focus of this work, suggested using models' confidence as an indication of answerability (Yin et al., 2023), and some suggested further finetuning to enhance the probability of detecting (un)answerable questions

---

[1]Our code is publicly available at https://github.com/lovodkin93/unanswerability

(Jiang et al., 2021; Kadavath et al., 2022). We, however, ask whether models already represent questions' (un)answerablility when producing answers, and find strong evidence for a positive answer.

Specifically, by experimenting with three QA datasets (Rajpurkar et al., 2018; Kwiatkowski et al., 2019; Trivedi et al., 2022), we observe a substantial increase in performance for (un)answerable questions (up to 80%) simply by incorporating to the prompt the possibility of (un)answerability. We further show that, even in the absence of guidance in the prompt, the fact that the question is (un)answerable is decodable from the model's representations. This is done by two methods: first, we find that the beam of decoded responses for (un)answerable queries often contains a response recognizing their (un)answerability; second, we demonstrate that the fact that the question is (un)answerable is easily decodable from the model's representations and that there is a *linear* separation between representations of (un)answerable and (un)answerable questions (see Figure 1). The existence of the *answerability subspace* is largely *independent* of the specific QA dataset used, in the sense that an answerability classifier trained over representations of questions from one dataset can successfully classify (un)answerable questions from other datasets as well. In addition to providing illuminating insights into the internal mechanics of LLMs, these findings also open up new avenues for better decoding methods (Meister et al., 2020; Wiher et al., 2022) to improve performance in general and on (un)answerable questions in particular.

## 2 Related Work

In previous research, (un)answerable questions were used to evaluate reasoning capabilities (Rajpurkar et al., 2018; Ferguson et al., 2020; Kwiatkowski et al., 2019; Trivedi et al., 2022). It was SQuAD v2 (Rajpurkar et al., 2018) that provided the first reading comprehension dataset for validating models' ability to deal with (un)answerability, by introducing questions that cannot be addressed from the given context. Kwiatkowski et al. (2019) followed the same line and included about a third of (un)answerable questions in their NATURAL QUESTIONS (NQ), an annotated open-domain QA dataset. Recently, MuSiQue (Trivedi et al., 2022) was introduced as a challenging multi-hop QA benchmark that consists

of (un)answerable questions, in which supporting paragraphs have been intentionally removed from the context. Our experiments use these datasets to demonstrate the effectiveness of our approach to identify (un)answerability.

(Un)answerability capabilities in LLMs were mainly studied by using few-shot prompting (Kandpal et al., 2022; Weller et al., 2023). Moreover, several works have recently shown that LLMs become easier to steer with natural language prompts either as they become larger (Mishra et al., 2022a; Kandpal et al., 2022; Carlini et al., 2023) or as they are exposed to larger instruction tuning data (Mishra et al., 2022b; Chung et al., 2022; Wan et al., 2023a), and as a consequence, it might improve the (un)answerability capabilities of the model. Specifically, in this work, we utilize prompt manipulation in order to systematically reveal to the model the option of avoiding answering hard questions. Automatic prompt tuning can be also used for improving (un)answerability capabilities, without the need for manual handcrafting prompts. Liao et al. (2022) introduced a prompt tuning-based strategy to mitigate (un)answerable questions, by mapping questions into their proper, specific templates.

Other works tried to manipulate the model predictions towards better (un)answerability via using data augmentation (Zhu et al., 2019), and Asai and Choi (2021) provided an in-depth analysis of the ability to detect (un)answerability in LMs, where the case study is the data which is fed to the model.

Furthermore, recent studies have suggested utilizing recent advances in white-box model interpretability (Geva et al., 2022b; Li et al., 2022; Mallen et al., 2022; Mickus et al., 2022; Meng et al., 2023; Geva et al., 2023) and probing (Adi et al., 2017; Conneau et al., 2018; Voita et al., 2019; Slobodkin et al., 2021) for manipulating the model predictions and analyzing when LLMs struggle to answer questions. Recent works also tried to use beam search decoding to manipulate the generated outputs by using the information encapsulated in several beams (Meister et al., 2020; Leblond et al., 2021; Slobodkin et al., 2023; Wan et al., 2023b). Finally, early exiting in language models (Schwartz et al., 2020; Schuster et al., 2022; Din et al., 2023) and model prediction calibration (Desai and Durrett, 2020; Jiang et al., 2021; Dhuliawala et al., 2022; Geva et al., 2022a) are strongly related to our work, as they suggest to analyze and improve the model predictions and output distribution.

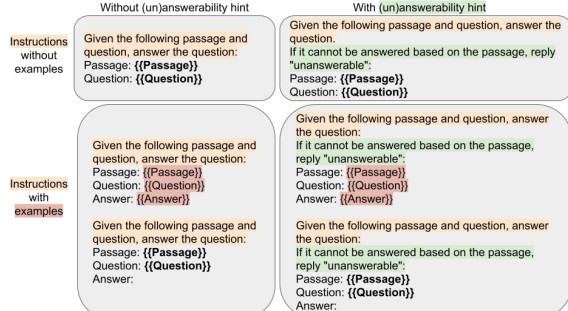

Figure 2: Combinations of prompt variants in this work. In addition to some basic instructions, our prompts can also have a "hint" to the possibility of (un)answerability, as well as 2 exemplars.

## 3 Method

We posit the hypothesis that, despite the inclination of LLMs to produce answers to (un)answerable queries, they do encode the (un)answerability of such queries within their latent representations. We examine this hypothesis by undertaking three distinct experimental approaches: (1) prompt manipulation, (2) beam scrutiny, and (3) probing (including identification and erasure of an answerability subspace).

### 3.1 Prompt Manipulation

First, we ask whether the model's ability to identify (un)answerable questions is sensitive to the exact wording of the prompt. Specifically, we ask whether merely raising the option of unasnwerability makes the model less susceptible to hallucination. To that end, we experiment with two types of prompts. The first type is designed to merely guide the model towards addressing a question. The second type, however, is more instructive in its approach. Besides guiding the model, it provides an advised course of action for scenarios where the question at hand is (un)answerable, hence indirectly hinting at the potential for (un)answerability.

Our experimental setup encompasses both zero-shot and few-shot prompts, with the latter involving the integration of two exemplars in the prompt. In the standard prompt setup, both exemplars are answerable. However, within the hinting prompt framework, one exemplar is designed to be (un)answerable. Figure 2 demonstrates all variants.

### 3.2 Beam Relaxation

Recall that the output of LMs is usually decoded with algorithms such as beam search. We aim to ex-

amine whether we can endow this algorithm with a bias towards unanswerability. Focusing on the zero-shot setting, we gradually increase the beam size. Then, instead of automatically choosing the highest-probability answer from the final set of $k$ options, we search for a reply within the final $k$ options that signifies (un)answerability (Appendix A). If such an answer is discovered, we substitute the top-beam answer with "unanswerable".

### 3.3 Identifying an Answerability Subspace

In a subsequent set of experiments, our objective is to find evidence for (un)answerability encoding directly in the embedding space of the models, by probing the models' last hidden layer. For each task, each model is prompted with a balanced trainset comprising 400 answerable and 400 (un)answerable examples. Then, for each instance, we take the embedding from the final hidden layer of the first generated token and train a linear classifier, using logistic regression, to predict answerability.[2] Subsequently, we assess the performance of each classifier on the corresponding test set. As a baseline, we also conduct similar experiments using the initial (non-contextual) embedding layer, which should not encode whether the question is answerable or not. Our core objective within this experimental setup is to ascertain whether a basic linear classifier, trained on a modestly sized dataset, suffices to effectively discriminate between answerable and (un)answerable queries.

### 3.4 Erasing the Answerability Subspace

Upon identifying a linear subspace that corresponds to (un)answerability, a natural question to ask is whether that subspace has a behavioral relevance, i.e., whether it is being *used* by the model when producing text. Importantly, this is different than mere encoding of the information, as the information can be present in the representation and at the same time be irrelevant to the model's behavior (Hewitt and Liang, 2019; Elazar et al., 2021; Ravfogel et al., 2021). Recent work on linear concept erasure (Ravfogel et al., 2020, 2022b; Belrose et al., 2023) have proposed a set of methods to erase arbitrary linearly-encoded concepts from neural representations, following the intuition that by erasing a subspace that encodes the concept and examining the effect on the model's output,

---

[2]We also experimented with averaging across all the generated tokens' embeddings, which was found to yield inferior performance on a designated development set.

|        | Answerable | (Un)answerable |
|--------|-----------|----------------|
| SQuAD  | 5928      | 5945           |
| NQ     | 3489      | 7719           |
| MuSiQue| 1950      | 1316           |

Table 1: Number of extracted answerable and (un)answerable questions per dataset in our test set.

we can verify that the subspace we identify is behaviorally meaningful, opening an avenue for performing interventions in that subspace in order to modify the model's behavior. These methods start with the original representations alongside binary labels (e.g., representations of the text alongside binary gender annotations for each text), and return a new representation which is linearly guarded in the sense that any linear classifier trying to recover the concept from the representation will fail. While linear erasure has its limitations (Ravfogel et al., 2022a), it has been proven to be an effective method for intervening in the latent representations of black-box models.

We use the recently proposed method of Belrose et al. (2023) which provides a closed-form solution for the concept-erasure objective. Concretely, given a binary concept (answerability), the method provides a projection matrix that minimally changes the representations (in the $L_2$ sense) while at the same time guarantees the inability to linearly predict the answerability from the modified representations. We fit the method over the last-layer representations of the training instances from Flan-UL2, particularly when these instances are prompted with regular queries from the SQuAD benchmark (refer to §3.3). Then, during inference, the concept-erasing projection matrix is applied in the first generation step, specifically for the test set within the same model-dataset pairing. Our goal is to inspect whether removing the linear separation that exists in the latent space of the model between answerable and (un)answerable questions changes the behavior of the model.

## 4 Experimental Setup

Our experiments focus on several language models and on three benchmarks.

**Benchmarks** We consider three QA benchmarks, incorporating (un)answerable questions, in a reading comprehension setting where models are tasked with responding to a question within a given context. For each benchmark, we use the entire development set to construct our testing dataset. Ad-

ditionally, for the probing experiments involving the training of linear classifiers on the models' embeddings, sample 1000 instances from each benchmark's trainset, evenly distributed between answerable and (un)answerable instances. Of these, we reserve 800 instances for the training of classifiers, with the remaining instances forming the development set for these classifiers. Below, we describe how we associate asnwerable questions with paragraphs that contain the answer, and how we associate (un)answerable questions with challenging paragraphs (that do not contain the answer, but may be topically similar to the question).

Our first benchmark is SQuAD 2.0 (Rajpurkar et al., 2018), a reading comprehension dataset, composed of manually-curated question-answer pairs alongside (un)answerable questions, each derived from a single paragraph.

The second benchmark we explore is NATURAL QUESTIONS (NQ; Kwiatkowski et al., 2019), a dataset accumulated from user-generated queries on the Google search engine. Each item within the dataset consists of a question, a retrieved article, a selected paragraph from the article (referred to as the "long answer"), and a short answer inferable from the paragraph. Despite its potential to test QA systems with a retrieval component, our interest lies exclusively in the question-answering setting, hence we utilize the "long answer" as the context, assuming an oracle retrieval system. For the formulation of answerable instances, we select cases with both a long and a short answer, using the former as the context and the latter as the response. For the (un)answerable questions, we pair each query with a paragraph from the sourced passage that has not been annotated as the "long answer". In order to create a challenging dataset, we select the paragraph that is closest in meaning to the question. To achieve this, we encode both the question and all potential paragraphs using a sentence-transformer (Sentence-Bert; Reimers and Gurevych, 2019) and select the paragraph that exhibits the highest cosine-similarity score.

Our final benchmark is the MuSiQue dataset (Trivedi et al., 2022), a multi-hop dataset featuring both answerable and (un)answerable questions. Each instance consists of a question, several candidate paragraphs, an answer, and a decomposition of the question into its single-hop sub-questions. Additionally, each sub-question is paired with a paragraph that has its answer, with all those align-

|            |         | SQuAD | NQ   | MuSiQue |
|------------|---------|-------|------|---------|
| Flan-T5$_{xxl}$ | Regular | 37.3  | 5.7  | 8.2     |
| (11B)      | +hint   | **91.5** | **85.6** | **74.7** |
| Flan-UL2   | Regular | 46.3  | 13.8 | 6.8     |
| (20B)      | +hint   | **92.3** | **83.9** | **67.3** |
| OPT-IML    | Regular | 43.9  | 21.8 | 17.1    |
| (30B)      | +hint   | **85.4** | **85.3** | **54.1** |

(a) Zero-shot

|            |         | SQuAD | NQ   | MuSiQue |
|------------|---------|-------|------|---------|
| Flan-T5$_{xxl}$ | Regular | 52.6
(11.7) | 8.6
(2.1) | 13.6
(4.9) |
| (11B)      | + hint  | **91.2**
**(1.0)** | **85.8**
**(0.5)** | **75.4**
**(1.4)** |
| Flan-UL2   | Regular | 67.7
(4.0) | 20.6
(1.8) | 14.5
(4.9) |
| (20B)      | + hint  | **92.5**
**(0.1)** | **83.7**
**(0.1)** | **72.1**
**(0.6)** |
| OPT-IML    | Regular | 10.0
(1.2) | 11.1
(2.1) | 4.9
(1.0) |
| (30B)      | + hint  | **79.3**
**(0.3)** | **85.0**
**(1.5)** | **27.5**
**(8.0)** |

(b) Few-shot

Table 2: F1 scores over the (un)answerability classification task in both zero-shot and few-shot setting. Each model is prompted with a regular prompt and with a prompt that hints at the possibility of (un)answerability ("+hint"). In the few-shot setting, results are averaged across three variations of in-context-learning examples (with standard deviation in brackets). **Bold** marks the better prompting method.

ing paragraphs concatenated and used as context. Conversely, for the (un)answerable queries, the absence of such alignment for some sub-questions is observed. For these (un)answerable instances, we identify the paragraph most closely linked to each of the unanswered single-hop questions, using a process akin to the approach with the NQ benchmark. These identified paragraphs are then aggregated, together with the paragraphs corresponding to the other single-hop questions, to form the context for the (un)answerable queries. Table 1 details the full statistics of our test sets across all three benchmarks. These test sets are obtained from the development set of each respective benchmark.

**Evaluation** The main task over which we evaluate models is the **(un)answerability classification task**. When evaluating QA models over this task we only examine whether they *tried* to answer, i.e., we count every example for which the model provides an answer as an instance of answerability prediction, and each example for which the

model did not provide an answer as an instance of un-answerability prediction,[3]. Note that this evaluation does not consider the correctness of the answers provided. The metric associated with this task is the F1 score, with "unanswerable" considered the positive label. Linear classifiers (§3.3) are also evaluated over the (un)answerability classification task, as the classifier is trained to predict whether or not the question is answerable, based on the hidden representations of the LM.

Additionally, in order to make sure that our methods do not hinder the performance of the models over their primary task, we evaluate them over the **QA task** as well, using the splits provided by the tasks' designers. We report the commonly used metrics: exact match (EM) and (token-wise) F1 scores (Rajpurkar et al., 2016).

**Language Models** We evaluate three instruction-finetuned large language models: namely, the 'xxl' variant of the Flan-T5 model (Flan-T5$_{xxl}$; Chung et al., 2022), the Flan-UL2 model (Chung et al., 2022), and the OPT-IML model (Iyer et al., 2023).

## 5 Results

### 5.1 Prompt Manipulation

#### 5.1.1 Zero-Shot Scenario

Table 2a presents the results in the zero-shot setting (the model was not provided with question-answer examples). It shows that the detection of (un)answerable questions is substantially improved upon the integration of a hint towards the possibility of (un)answerability into the prompts, with gains as high as 80 points. It can also be observed that, without a hint, the ability to discern (un)answerable queries tends to be superior in larger models. Interestingly, the introduction of the hint appears to mitigate the impact of model size, as evidenced by the smaller Flan-T5$_{xxl}$ surpassing its larger counterparts in two out of three benchmark evaluations.

Additionally, Table 3a displays the models' exact match and token-wise F1 scores over the QA task where the model is tasked with both detecting (un)answerable questions and provide a correct answer to the answerable ones. It reveals a notable enhancement in the quality of generated responses when prompted with a hint, in some cases resulting in improvements of over 50 points (on both met-

---

[3]After analyzing the models' responses, we curated a list of answers that signify abstaining from answering. See Appendix A for further details.

|  |  | SQuAD | | NQ | | MuSiQue | |
|---|---|---|---|---|---|---|---|
|  |  | EM | F1 | EM | F1 | EM | F1 |
| Flan-T5$_{xxl}$ | Regular | 55.4 | 58.6 | 22.5 | 27.0 | 41.2 | 48.5 |
| (11B) | +hint | **86.5** | **89.6** | **73.3** | **77.0** | **63.5** | **70.5** |
| Flan-UL2 | Regular | 59.5 | 62.3 | 21.7 | 27.3 | 35.3 | 43.4 |
| (20B) | +hint | **87.8** | **90.5** | **70.5** | **74.5** | **55.3** | **62.2** |
| OPT-IML | Regular | 57.8 | 60.6 | 29.0 | 33.0 | 37.3 | 44.4 |
| (30B) | +hint | **81.2** | **83.5** | **73.9** | **76.7** | **47.7** | **54.5** |

(a) Zero-shot

|  |  | SQuAD | | NQ | | MuSiQue | |
|---|---|---|---|---|---|---|---|
|  |  | EM | F1 | EM | F1 | EM | F1 |
| Flan-T5$_{xxl}$ (11B) | Regular | 61.8 (6.1) | 65.2 (5.7) | 23.8 (1.4) | 28.2 (1.3) | 41.3 (2.9) | 48.9 (2.9) |
|  | + hint | **86.0 (1.6)** | **89.2 (1.4)** | **73.6 (0.4)** | **77.3 (0.3)** | **62.9 (2.7)** | **69.9 (2.6)** |
| Flan-UL2 (20B) | Regular | 69.8 (2.0) | 73.0 (2.2) | 26.1 (1.5) | 31.2 (1.4) | 40.0 (3.4) | 47.5 (3.0) |
|  | + hint | **87.9 (0.4)** | **90.7 (0.3)** | **70.7 (0.2)** | **74.9 (0.2)** | **60.1 (0.2)** | **67.4 (0.2)** |
| OPT-IML (30B) | Regular | 44.2 (0.6) | 47.5 (0.6) | 24.5 (0.7) | 28.4 (0.7) | 31.7 (1.3) | 39.2 (1.4) |
|  | + hint | **74.5 (0.5)** | **76.6 (0.8)** | **73.6 (2.2)** | **75.9 (1.9)** | **36.7 (2.3)** | **44.3 (2.3)** |

(b) Few-shot

Table 3: Exact match (EM) and (token) F1 scores over the QA task in zero-shot and few-shot setting. For each model, there are two prompt variants: regular and with a hint of the possibility of (un)answerability. In the few-shot setting, results are averaged across three variations of in-context-learning examples (with standard deviation in brackets). **Bold** marks the better prompting method.

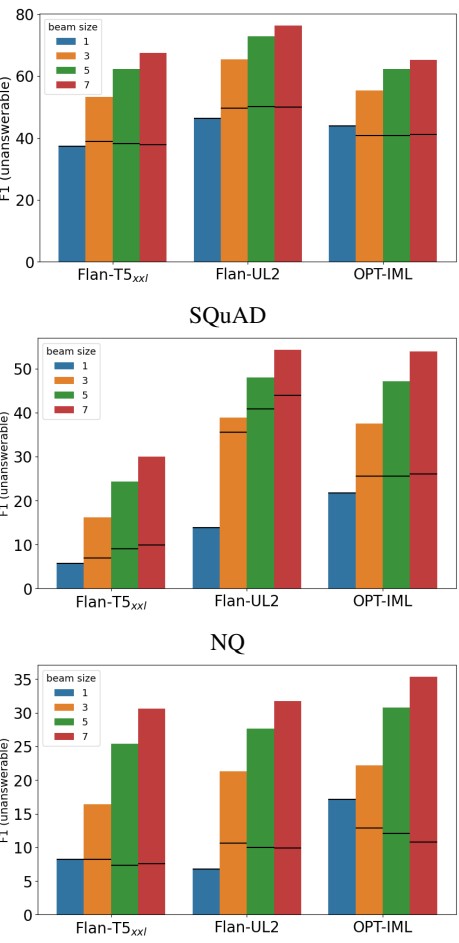

Figure 3: F1 over the (un)answerability classification task with beam relaxation. In this setting, the models were considered successful iff a reply acknowledging the (un)answerability of a question was found anywhere in the beam. The horizontal lines show the F1 for the usual metric, i.e., successful classification only if the correct reply was on the top of the beam.

rics). The improvement over the QA task arises in large part from models giving the correct response to (un)answerable questions. We observe an average drop of 8.3% in F1 and 7.1% in exact match over answerable questions in a zero-shot setting when providing the hint (see Appendix B for all the results over answerable questions).

### 5.1.2 Few-Shot Scenario

Table 2b provides an exhaustive overview of the results in the few-shot setting. In order to mitigate the impact of the chosen examples, we experiment with three variants of in-context examples for each benchmark, and report the average results, as well as the standard deviation.[4] Mirroring the trend seen in the zero-shot scenario, when the prompts encapsulate a hint towards the potential of (un)answerability, there is a significant improvement in the identification of (un)answerable queries. This trend is further corroborated by Table 3b, which reports the exact match and token-wise F1 scores of the models over the QA task. See

Appendix D for a comparison of the two possible hints: in the instructions, and as an (un)answerable example.

### 5.2 Beam Relaxation

Figure 3 illustrates the models' ability to detect (un)answerable queries, when gradually increasing the beam size. Although the increase in beam size yields a negligible impact on the final, most probable response (as depicted by the horizontal lines in Figure 3), it shows better recognition of (un)answerability. This is illustrated by a consistent increase in the presence of (un)answerability-acknowledging responses[5] within one of the beams

---

[4]See Appendix F for further details.

[5]Please refer to Appendix A for a comprehensive overview of such responses.

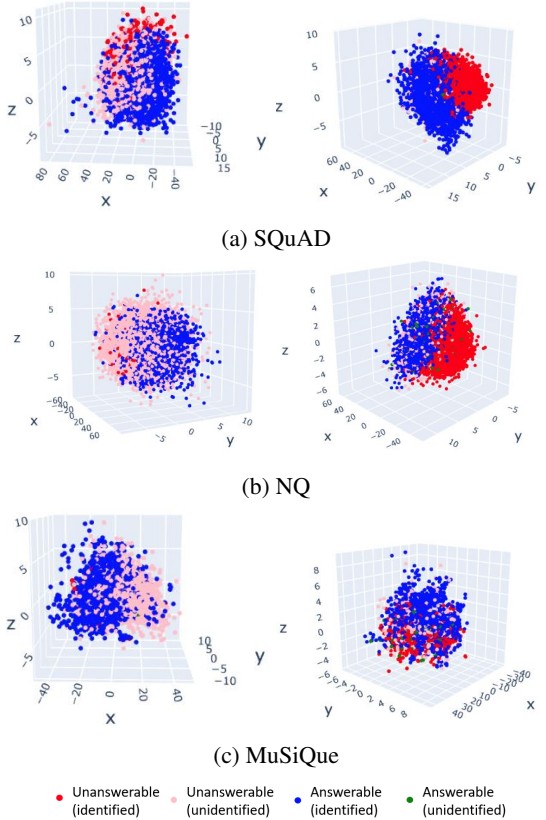

(a) SQuAD

(b) NQ

(c) MuSiQue

● Unanswerable  ● Unanswerable  ● Answerable  ● Answerable
  (identified)      (unidentified)    (identified)     (unidentified)

Figure 4: 3D PCA projection of the last hidden layer's embedding of the Flan-T5$_{xxl}$ model on each of the three benchmarks. The left images show the embeddings with the regular prompt, and the right ones - with the prompt with a hint.

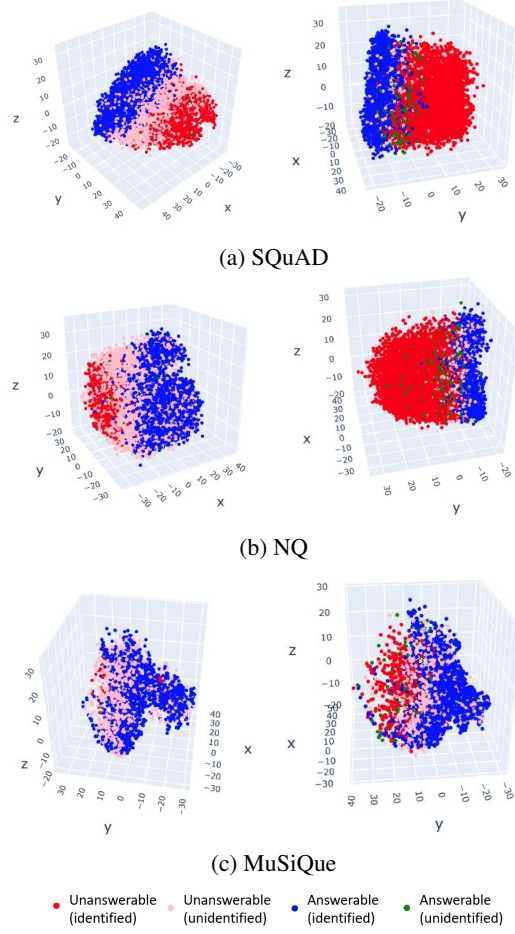

(a) SQuAD

(b) NQ

(c) MuSiQue

● Unanswerable  ● Unanswerable  ● Answerable  ● Answerable
  (identified)      (unidentified)    (identified)     (unidentified)

Figure 5: 3-D PCA projection of the last hidden layer's embedding of the OPT-IML model on each of the three benchmarks. The left images show the embeddings with the regular prompt, and the right ones - with the prompt with a hint.

(signified by the height of the bars). This observation underscores the notion that beneath the facade of overconfidence expressed by these models, the models do encode their inability to respond to certain queries. Importantly, we find that this approach has very little negative impact on the *answerable* questions, with only a slight degradation in the exact match and F1 scores (see Appendix E for further details).

Notably, we conjecture that the observed decrease in performance on the NQ and MuSiQue benchmarks, compared to SQuAD, can be attributed to two main factors: distribution shift and a more challenging task environment. One contributing factor is the non-conventional format of queries in NQ; unlike the typical question format found in datasets like SQuAD, NQ queries do not always adhere to this pattern. Language models (LLMs) primarily trained on question-answering datasets, like SQuAD, might struggle with this distribution shift, leading to a decline in their performance when faced with non-question-formatted

queries.

In addition, the MuSiQue dataset introduces a significant challenge by requiring multi-hop reasoning. There are limited datasets available on which models can be trained for such complex tasks, and even fewer with (un)answerable questions. This scarcity, coupled with the demand for multi-hop reasoning, amplifies the difficulty of MuSiQue. This high complexity is further highlighted by the diminished performance of models, even when responding to answerable questions, as evident in Tables 6 and 9 in Appendices B and E, respectively. This drop in performance shows how challenging these benchmarks are, especially when compared to easier ones like SQuAD.

### 5.3 Identifying an Answertability Subspace

We report the performance of the linear classifiers in Table 4. Notably, when considering the stan-

| Model | | SQuAD | | NQ | | MuSiQue | |
|---|---|---|---|---|---|---|---|
| | | $1^{st}$ layer | last layer | $1^{st}$ layer | last layer | $1^{st}$ layer | last layer |
| Flan-T5$_{xxl}$ | regular | 40.1 | 89.9 | 23.0 | 86.1 | 47.4 | 77.5 |
| (11B) | +hint | 40.0 | 89.4 | 26.6 | 86.2 | 38.6 | 77.3 |
| Flan-UL2 | regular | 39.4 | 90.4 | 42.2 | 87.3 | 15.1 | 78.3 |
| (20B) | +hint | 39.6 | 89.9 | 41.5 | 87.9 | 41.6 | 78.3 |
| OPT-IML | regular | 48.4 | 82.8 | 40.8 | 85.5 | 45.6 | 75.5 |
| (30B) | +hint | 48.4 | 83.9 | 45.3 | 86.2 | 45.6 | 84.9 |

Table 4: F1 scores of (un)answerability classification of the linear classifier trained for each model-dataset pair, once with the regular prompt and once with the prompt that hints at the possibility of (un)answerability. For each model, we classify once based on the first layer and once based on the last layer of the first generated token.

dard prompt, the F1 of the probe is above 75% for all models and datasets. Furthermore, we find that hinting to the possibility of (un)answerability only marginally improves the ability to correctly classify queries from the representations within the models. These suggest the existence of an '(un)answerability' linear subspace.

**Visuazliation.** To examine this hypothesis, we perform a PCA projection of the embedding of the final hidden layer of the first generated token onto a 3-D plane. Figures 1, 4, 5 display the results for the Flan-UL2, Flan-T5$_{xxl}$ and OPT-IML models, respectively. Consistent with our hypothesis, it can be observed that (un)answerable queries, which were correctly identified as such by the model (depicted by red dots in the figures), are distinctly separate from the answerable queries (represented by blue dots in the figures). This separation becomes especially pronounced in the context where the prompt incorporated a hint (as illustrated in the right subfigures). Importantly, we find that (un)answerable questions, which the models failed to recognize as such and instead generated a hallucinated response (indicated by pink dots in the figures), appear to reside within a separate linear subspace. This finding demonstrates that, notwithstanding the overconfidence exhibited by these models, they intrinsically possess the capacity to distinguish (un)answerable queries. This intrinsic capability is particularly evident given that the subspace corresponding to hallucinated (un)answerable questions (pink) seems to be positioned between that of the answerable queries (blue) and that of correctly identified (un)answerable queries (red). This positioning is suggestive of the models' inherent uncertainty.

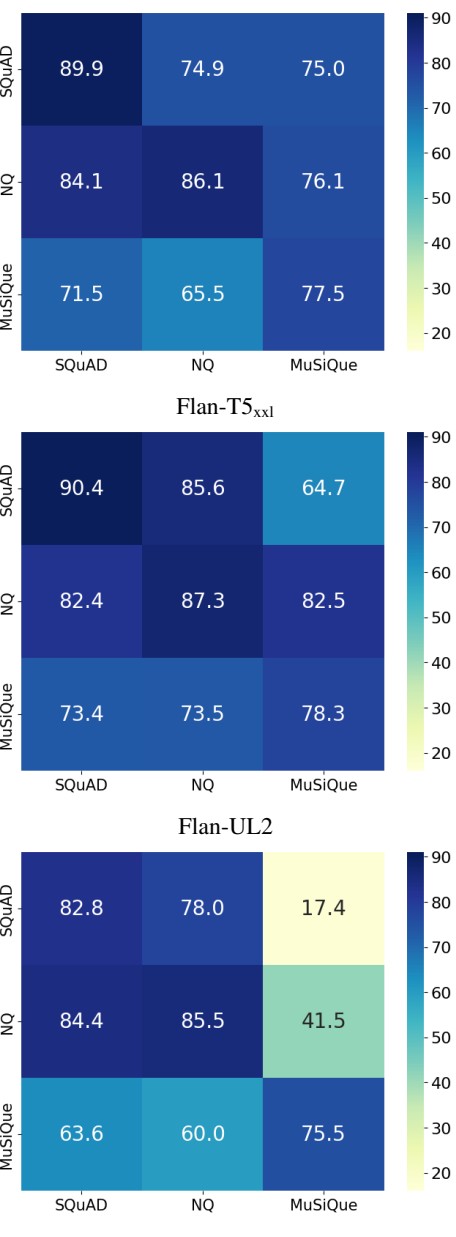

Figure 6: F1 scores of (un)answerability classification, as determined by a linear classifier trained for each model-dataset pair, and tested on the other benchmarks. Within each heatmap, the column designates the dataset used for training, while the row illustrates the dataset on which the classifier was tested.

**Trasnfer Between Datasets.** In Figure 6 we present the transferability of the (un)answerability classifier trained on a given dataset to other datasets. While performance deteriorates, the F1 scores are still well above the F1 scores we calculated over the uncontextualized first layer. This suggests that, to a large degree, the probes identify an abstract (un)answerability subspace beyond dataset-specific shallow features.

| k-beam Type | | All | | Answerable | |
|---|---|---|---|---|---|
| | | EM | F1 | EM | F1 |
| Regular | w\o erasure | 60.2 | 63.8 | 87.1 | 94.1 |
| | with erasure | 50.2 | 55.1 | 82.0 | 91.8 |
| Relaxed | w\o erasure | 60.9 | 64.4 | 87.1 | 94.1 |
| | with erasure | 50.8 | 55.7 | 82.0 | 91.8 |

Table 5: Exact match (EM) and F1 scores of all questions and of answerable questions in the zero-shot setting for the Flan-UL2 model on the SQuAD benchmark with a beam size of 3. The results demonstrate the performance before and after the application of the concept erasure, for the regular k-beam decoding approach, and the relaxed variant.

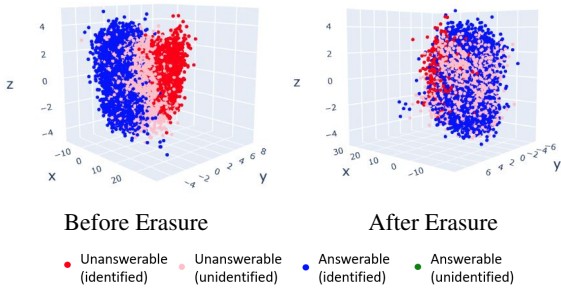

Figure 7: 3-D PCA projection of the last hidden layer's embedding of the Flan-UL2 model on the SQuAD dataset, without performing erasure (left) and after (right).

## 5.4 Erasing the Answerability Subspace

Recall that if the subspaces we found are causally related to the predictions of the model, we expect that erasing them would deteriorate the model's performance in the answerability task. Indeed, when linearly erasing the answerability subspace from the first token representation of Flan-UL2, we see the F1 score over the (un)answerability classification task decreasing from 50.1 to 31.2 with regular beam, and from 65.4 to 32.7 with beam relaxation. This trend is also evident from the results on the QA task presented in Table 5, as well as when projecting the embeddings on a 3-D plane using PCA, as depicted in Figure 7. This suggests that the answerability subspace is influencing the model's behavior in the context of the answerability task.

## 6 Conclusion

We found ample evidence for LM's ability to encode the (un)answerability of questions, despite the fact that models tend to be over-confident and generate hallucinatory answers when presented with (un)answerable questions. We also showed that this discrepancy between model output and its hidden states is mitigated by simply adding the option of (un)answerability to the prompt.

The evidence we found includes the existence of a reply acknowledging the (un)answerability in a beam of decoded answers, meaning that even though the models' best-assessed answer is hallucinatory, the true answer is not lagging too far behind. We also showed that the models' representations after encoding the question and before decoding the answer are highly influenced by the answerability of the question or lack thereof, with answerable and (un)answerable questions being linearly separable in the embedding space. We conclude that

the problem of answered (un)answerable questions can be mended with either better prompting, better decoding, or simple auxiliary classification models.

## 7 Limitations

We focus on a few datasets and models. Despite the effort to experiment with several models, future work should experiment with different models, and in particular, examine the relation between the ability to encode (un)answerability and model scale. We also do not compare the different approaches explored in this paper, which we leave as an interesting future research direction. Our focus on linear probing and linear erasure stems from the availability of existing methods from this family, but deep LMs are highly nonlinear and may encode the information we are interested in in a nonlinear manner. As such, our results should only be interpreted as a lower bound for the identification of (un)answerability. Lastly, our experiments focused on (un)answerability in a given context. Future work should also explore the phenomenon in the open-domain setting.

## 8 Ethics Statement

Model hallucination, in general, can have real-world implications when models are incorporated in, e.g., search engines or other applications. Our study focuses on the ability to discern a specific type of hallucination in a selected set of models and datasets. It should not be taken as a general solution to the problem of hallucination in the QA setting, but rather as preliminary research on potential techniques for mitigating the problem of hallucination.

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

## A  (Un)answerability-Recognizing Responses

After analyzing the responses generated by the different models in this work, we curated a list of answers that signify abstention from answering, which we used to identify responses that signify (un)answerability. This includes: "Unanswerable", "N/A", "I don't know", "IDK", "Not known", "Answer not in context", "Unknown", "No answer", "It is unknown", "None of the above", "None of the above choices", "The answer is unknown", along with their corresponding versions in lowercase.

## B  Performance on the Answerable Instances

Table 6 show the exact match and F1 scores of each model over the QA task only on the answerable questions of each benchmark, in the zero-shot setting and the few-shot setting. Note that although the addition of the hint hinders the models' performance, the drop over the answerable questions is small for the most part and outbalanced by the improvement of detection of (un)answerable questions, leading to the overall improvement shown in Table 3.

## C  Prompt Variant Tuning

In our work, we experiment with three variants of the prompt containing a hint of the possibility of (un)answerability. These are:

1. Given the following passage and question, answer the question. If it cannot be answered based on the passage, reply "unanswerable".

2. Given the following passage and question, answer the question. If you don't know the answer, reply "IDK".

3. Given the following passage and question, answer the question. If there is no correct answer, reply "N/A".

We run all three variants on a separate development set[6], to decide which prompt to use for each model and dataset. Table 7 shows the results of all three variants on our development set. Based on these results, we decide to use the first variant in our experiments on the SQuAD and NQ datasets

---

[6]The development set consists of 200 answerable and 200 (un)answerable instances, extracted from the train set of each respective benchmark.

|  |  | SQuAD | | NQ | | MuSiQue | |
|---|---|---|---|---|---|---|---|
|  |  | EM | F1 | EM | F1 | EM | F1 |
| Flan-T5$_{xxl}$ | Regular | **88.0** | **94.3** | **65.5** | **80.0** | **66.1** | **78.4** |
| (11B) | +hint | 86.2 | 92.3 | 56.3 | 68.2 | 61.2 | 72.9 |
| Flan-UL2 | Regular | **89.0** | **94.7** | 53.1 | **71.2** | **56.7** | **70.1** |
| (20B) | +hint | 87.8 | 93.2 | **54.3** | 67.0 | 52.1 | 63.6 |
| OPT-IML | Regular | **87.6** | **93.2** | **66.1** | **78.9** | **56.1** | **67.9** |
| (30B) | +hint | 80.2 | 84.9 | 53.9 | 63.0 | 52.0 | 63.3 |

(a) Zero-shot

|  |  | SQuAD | | NQ | | MuSiQue | |
|---|---|---|---|---|---|---|---|
|  |  | EM | F1 | EM | F1 | EM | F1 |
| Flan-T5$_{xxl}$ | Regular | **87.4** | **94.2** | **66.3** | **80.3** | **64.2** | **76.9** |
| (11B) |  | **(1.9)** | **(1.1)** | **(1.9)** | **(1.6)** | **(3.1)** | **(3.0)** |
|  | + hint | 84.6 | 91.1 | 55.5 | 67.3 | 57.4 | 69.1 |
|  |  | (2.8) | (2.4) | (5.1) | (5.5) | (5.4) | (5.3) |
| Flan-UL2 | Regular | **88.3** | **94.6** | 58.3 | **74.7** | **61.7** | **74.2** |
| (20B) |  | **(0.7)** | **(0.3)** | (2.5) | **(2.1)** | **(3.8)** | **(3.2)** |
|  | + hint | 87.1 | 92.7 | **59.0** | 72.4 | 56.4 | 68.6 |
|  |  | (1.0) | (0.6) | **(0.1)** | (0.2) | (0.8) | (0.7) |
| OPT-IML | Regular | **83.0** | **89.6** | **65.2** | **77.8** | **51.4** | **63.9** |
| (30B) |  | **(0.5)** | **(0.4)** | **(0.7)** | **(0.8)** | **(1.8)** | **(2.0)** |
|  | + hint | 69.6 | 73.8 | 48.8 | 56.3 | 50.3 | 63.0 |
|  |  | (5.0) | (5.7) | (3.2) | (4.3) | (1.3) | (1.2) |

(b) Few-shot

Table 6: Exact match (EM) and F1 scores over the QA task only for answerable questions in zero-shot and zero-shot setting. For each model, there are two prompt variants: regular and with a hint of the (un)answerability. In the few-shot setting, results are averaged across three variations of in-context-learning examples (with standard deviation in brackets). **Bold** marks the better prompting method.

| Model | Prompt Variant | SQuAD | | NQ | | MuSiQue | |
|---|---|---|---|---|---|---|---|
|  |  | Exact | F1 | Exact | F1 | Exact | F1 |
| Flan-T5-xxl | 1 | 85.0 | 89.4 | 63.0 | 71.5 | 59.0 | 66.1 |
| (11B) | 2 | 60.5 | 67.2 | 52.5 | 61.2 | 37.5 | 44.9 |
|  | 3 | 78.0 | 84.2 | 61.5 | 69.6 | 67.0 | 73.1 |
| Flan-UL2 | 1 | 84.0 | 90.0 | 60.5 | 69.0 | 56.0 | 63.5 |
| (20B) | 2 | 74.5 | 80.5 | 55.0 | 63.6 | 48.5 | 56.2 |
|  | 3 | 80.5 | 86.8 | 59.5 | 67.4 | 63.0 | 69.5 |
| OPT-IML | 1 | 82.0 | 86.6 | 69.0 | 74.2 | 46.5 | 52.9 |
| (30B) | 2 | 59.0 | 65.2 | 52.5 | 58.3 | 39.5 | 46.3 |
|  | 3 | 65.5 | 71.6 | 57.5 | 63.5 | 60.0 | 63.9 |

Table 7: Exact match and (token) F1 scores in the zero-shot setting for three variants of the prompt containing a hint of the possibility of (un)answerability, on our development set.

(on all models), and the third variant in our experiments on the MuSiQue dataset (on all models).

## D  Impact of Hint Placement

To gain a deeper understanding of the circumstances under which the addition of a hint in the few-shot scenario is most beneficial, we conduct two ablations on the prompts. In one we gave a hint only in the instructions with all examples answerable, while in the other we gave special instructions but one of the two examples was (un)answerable.

|  |  | SQuAD | NQ | MuSiQue |
|---|---|---|---|---|
| Flan-T5$_{xxl}$ (11B) | Regular | 52.6 (11.7) | 8.6 (2.1) | 13.6 (4.9) |
|  | +hint (E&I) | **91.2** **(1.0)** | **85.8** **(0.5)** | **75.4** **(1.4)** |
|  | +hint (I) | 91.1 (1.0) | 85.0 (0.2) | 74.2 (1.3) |
|  | +hint (E) | 88.7 (2.1) | 81.9 (0.8) | 51.3 (7.2) |
| Flan-UL2 (20B) | Regular | 67.7 (4.0) | 20.6 (1.8) | 14.5 (4.9) |
|  | +hint (E&I) | **92.5** **(0.1)** | **83.7** **(0.1)** | **72.1** **(0.6)** |
|  | +hint (I) | 92.3 (0.1) | 82.3 (0.1) | 71.3 (1.0) |
|  | +hint (E) | 92.0 (0.1) | 79.7 (0.6) | 59.0 (1.7) |
| OPT-IML (30B) | Regular | 10.0 (1.2) | 11.1 (2.1) | 4.9 (1.0) |
|  | +hint (E&I) | **79.3** **(0.3)** | **85.0** **(1.5)** | **27.5** **(8.0)** |
|  | +hint (I) | 75.0 (1.8) | 81.9 (1.1) | 14.1 (4.2) |
|  | +hint (E) | 58.1 (1.4) | 61.0 (6.9) | 8.4 (3.7) |

Table 8: F1 scores over the (un)answerability classification task in few-shot setting. Each model is prompted with a regular prompt, and with three types of hint-including prompts: only in the instructions ("+hint (I)"), only in the exemplars ("+hint (E)") and in both ("+hint (E&I)"). Results are averaged across three variations of in-context-learning examples (with standard deviation in brackets). **Bold** marks the better prompting method.

As per the data presented in Table 8, it is evident that the inclusion of a hint within the instructions is considerably more advantageous compared to its addition within the exemplars. Indeed, once a hint is incorporated within the instructions, further inclusion within the exemplars has a minimal impact on the results, with OPT-IML being the sole exception.

## E  Impact of the Relaxed Beam Search Decoding on the Answerable Queries

Table 9 associated exclusively with each model on answerable questions, evaluated under the framework of our beam inspection experiments (see Section §3.2). Two decoding approaches were employed: a conventional beam-search decoding and its relaxed variant where the top answers are supplanted by an (un)answerability-recognizing response if it emerges within the beam. Our findings suggest a marginal impact of the adapted beam-search on answerable queries, with a maximal reduction of 7.7 and 8.7 points observed in the exact match and F1 scores respectively, when compared

to its regular counterpart. Like in Appendix B, these results point to the fact that any improvement achieved in §5.2 indeed stems from better treatment of (un)answerable questions.

## F  In-Context-Learning Variants

In order to mitigate the effect of the chosen in-context examples in the few-shot setting, we experiment with 3 variants of in-context examples, and average their scores. Figure 8, Figure 9, and Figure 10 show the different in-context examples variants for the SQuAD, NQ, and MuSiQue tasks, respectively.

| Model | Beam size | SQuAD | | NQ | | MuSiQue | |
|---|---|---|---|---|---|---|---|
| | | Regular | Relaxed | Regular | Relaxed | Regular | Relaxed |
| Flan-T5$_{xxl}$ | 1 | 88.0 | 88.0 | 65.5 | 65.5 | 66.1 | 66.1 |
| (11B) | 3 | 81.0 | 80.7 | 60.5 | 59.3 | 59.8 | 58.9 |
| | 5 | 81.1 | 80.4 | 59.6 | 57.1 | 60.3 | 58.2 |
| | 7 | 81.8 | 80.5 | 60.4 | 56.3 | 61.2 | 58.6 |
| Flan-UL2 | 1 | 89.0 | 89.0 | 53.1 | 53.1 | 56.7 | 56.7 |
| (20B) | 3 | 79.4 | 77.4 | 48.4 | 47.8 | 53.4 | 49.1 |
| | 5 | 79.2 | 76.6 | 48.0 | 46.0 | 55.8 | 50.7 |
| | 7 | 79.7 | 75.9 | 48.1 | 44.4 | 57.3 | 51.4 |
| OPT-IML | 1 | 87.6 | 87.6 | 66.1 | 66.1 | 56.1 | 56.1 |
| (30B) | 3 | 82.4 | 79.4 | 64.7 | 63.0 | 50.6 | 48.0 |
| | 5 | 83.0 | 76.8 | 64.6 | 60.2 | 51.3 | 46.5 |
| | 7 | 83.6 | 78.6 | 64.4 | 56.8 | 52.4 | 44.6 |

(a) Exact Match

| Model | Beam size | SQuAD | | NQ | | MuSiQue | |
|---|---|---|---|---|---|---|---|
| | | Regular | Relaxed | Regular | Relaxed | Regular | Relaxed |
| Flan-T5$_{xxl}$ | 1 | 94.3 | 94.3 | 80.0 | 80.0 | 78.4 | 78.4 |
| (11B) | 3 | 91.4 | 91.1 | 77.6 | 76.3 | 74.3 | 73.3 |
| | 5 | 91.6 | 90.8 | 76.9 | 74.1 | 74.9 | 72.6 |
| | 7 | 91.7 | 90.3 | 77.3 | 72.7 | 75.0 | 71.9 |
| Flan-UL2 | 1 | 94.7 | 94.7 | 71.2 | 71.2 | 70.1 | 70.1 |
| (20B) | 3 | 90.4 | 88.3 | 67.2 | 66.6 | 70.7 | 65.8 |
| | 5 | 90.3 | 87.5 | 67.1 | 64.7 | 72.2 | 65.4 |
| | 7 | 90.6 | 86.4 | 66.8 | 62.2 | 72.9 | 65.5 |
| OPT-IML | 1 | 93.2 | 93.2 | 78.9 | 78.9 | 67.9 | 67.9 |
| (30B) | 3 | 90.3 | 87.3 | 77.3 | 75.5 | 63.0 | 60.2 |
| | 5 | 90.6 | 84.2 | 76.9 | 72.0 | 63.3 | 57.8 |
| | 7 | 91.0 | 85.7 | 76.7 | 68.1 | 64.2 | 55.5 |

(b) F1

Table 9: Exact match (top table) and F1 (bottom table) scores of answerable questions in the zero-shot setting for different beam sizes. The results demonstrate the performance of two decoding approaches: the conventional beam-search method ("regular"), and a modified relaxed beam-search variant ("w\textbackslash relax."). In the latter technique, the highest-ranking response is substituted by an (un)answerability-recognizing response, in cases where such a response is present within the breadth of the beam.

Variant1

Variant2

Variant3

Figure 8: The three variants of in-context examples for the SQuAD prompts. For the regular prompts, we use the two answerable examples, whereas for the prompts hinting at the possibility of (un)answerability, we use the first answerable example and the (un)answerable examples. Additionally, for the other prompt variants, we replace the "unanswerable" answer of the (un)answerable example with "IDK" and "N/A", accordingly.

Variant1

Variant2

Variant3

Figure 9: The three variants of in-context examples for the NQ prompts. For the regular prompts, we use the two answerable examples, whereas for the prompts hinting at the possibility of (un)answerability, we use the first answerable example and the (un)answerable examples. Additionally, for the other prompt variants, we replace the "unanswerable" answer of the (un)answerable example with "IDK" and "N/A", accordingly.

```
Answerable Example1:
Passage: Paragraph 1: South Africa have played at six of the eight Rugby World Cup tournaments, having been unable to compete in the first two
        tournaments due to a sports boycott during the apartheid era. Following the end of apartheid, they hosted the 1995 Rugby World Cup and won
        the tournament. Paragraph 2: With two tournament wins, South Africa is one of the three best performing teams, along with Australia who
        have also won twice, and New Zealand with three wins, the only team to do better.
Question: How many times did the winner of the 1995 Rugby World Cup win in total?
Answer: two times.
Answerable Example2:
Passage: Paragraph 1: Barack Obama is an American politician who served as the 44th president of the United States from 2009 to 2017. Pargraph
        2: Obama married Michelle on October 3, 1992, after being engaged for almost a year. Paragraph 3: Barack Obama was born in Honolulu,
        Hawaii. After graduating from Columbia University in 1983, he worked as a community organizer in Chicago.
Question: What is the name of the wife of the American president who was born in Hawaii?
Answer: Michelle.
Un-answerable Example:
Passage: Paragraph 1: Barack Obama is an American politician who served as the 44th president of the United States from 2009 to 2017. Pargraph
        2: Obama married Michelle on October 3, 1992, after being engaged for almost a year.
Question: What is the name of the wife of the American president who was born in New York?
Answer: unanswerable.
```

Variant1

```
Answerable Example1:
Passage: Paragraph 1: Kaya toast is a well-known snack in Singapore. Kaya toast is prepared with kaya (coconut jam), a topping of sugar, coconut
        milk and eggs, pandan, and sometimes margarine or butter. Kaya is generally served on toast, and also sometimes on crackers. It is
        considered a breakfast staple, and remains popular in Singapore. The dish is sometimes dipped into soft-boiled egg with a little dark soy
        sauce and white pepper. Paragraph 2: A justice of the peace in Singapore derives his powers from statute law. He is appointed by the
        President of the Republic of Singapore, under the provisions of section 11 (l) of the Subordinate Courts Act (Cap. 321). The President may
        revoke the appointment of any justice of the peace. A newly appointed justice of the peace is required by section 17 of the Subordinate
        Courts Act, to take the oath of office and allegiance as set out in the schedule to the Subordinate Courts Act, before exercising the
        functions of his office.
Question: How do you become a justice of peace in the country where Kaya toast is popular?
Answer: appointed by the President of the Republic of Singapore.
Answerable Example2:
Passage: Paragraph 1: Mount Henry is located in the Lewis Range, Glacier National Park in the U.S. state of Montana. Mount Henry is just south of
        Appistoki Peak in the Two Medicine region of the park. Paragraph 2: KJRZ-LP (105.3 FM) was a radio station in Libby, Montana. It was owned
        and operated by the Libby Area Chamber of Commerce. Paragraph 3: The Lewis Range is a mountain range located in the Rocky Mountains of
        northern Montana, United States and extreme southern Alberta, Canada. It was formed as a result of the Lewis Overthrust, a geologic thrust
        fault resulted in the overlying of younger Cretaceous rocks by older Proterozoic rocks. The range is located within Waterton Lakes National
        Park in Alberta, Canada and Glacier National Park and the Bob Marshall Wilderness Complex in Montana, United States. The highest peak is
        Mount Cleveland at .
Question: In what mountain group is the range of which Mount Henry from the state where KJRZ-LP is located is part?
Answer: Rocky Mountains.
Un-answerable Example:
Passage: Paragraph 1: WODS (103.3 MHz) - known on-air as 103.3 AMP Radio - is a commercial FM radio station in Boston, Massachusetts. WODS airs a
        Top 40 (CHR) radio format, and is owned by Entercom. Its studios and offices are located on Leo M. Birmingham Parkway in Brighton.
        Paragraph 2: The Embassy of the United States to the Republic of Indonesia is located in Jakarta just south of the Monas at Jalan Medan
        Merdeka Selatan. Paragraph 3: Westminster College is a private liberal arts college located in the Sugar House neighborhood of Salt Lake
        City, Utah, United States. The college comprises four schools: the School of Arts and Sciences, the Bill and Vieve Gore School of Business,
        the School of Education, and the School of Nursing and Health Sciences. It is the only accredited liberal arts college in the state of
        Utah. Paragraph 4: The Shorter House is located at the end of Andrews Road in Thompson Ridge, a hamlet in the Town of Crawford in Orange
        County, New York, United States. It is a late 18th-century building later modified in the Greek Revival style.
Question: What is the business category of Crawford House, located in the same city as WODS and the same state as Wellesley College in Mona Lisa
        Smile?
Answer: unanswerable.
```

Variant2

```
Answerable Example1:
Passage: Paragraph 1: Meet Me in St. Louis is a musical film made by Metro - Goldwyn - Mayer and released in 1944. Divided into a series of
        seasonal vignettes, starting with Summer 1903, it relates the story of a year in the life of the Smith family in St. Louis, leading up to
        the opening of the Louisiana Purchase Exposition (more commonly referred to as the World's Fair) in the spring of 1904. The picture stars
        Judy Garland, Margaret O'Brien, Mary Astor, Lucille Bremer, Tom Drake, Leon Ames, Marjorie Main, June Lockhart, and Joan Carroll.
        Paragraph 2: Gracie is a 2007 American sports drama film directed by Davis Guggenheim. It stars Carly Schroeder as Gracie Bowen, Dermot
        Mulroney as Bryan Bowen, Elisabeth Shue as Lindsay Bowen, Jesse Lee Soffer as Johnny Bowen, and Andrew Shue as Coach Owen Clark. Paragraph
        3: He was born Philip Davis Guggenheim in St. Louis, Missouri, United States, the son of Marion Davis and film director and producer
        Charles Guggenheim. His father was Jewish, whereas his mother was Episcopalian. He graduated from the Potomac School (McLean, Virginia)
        (1979), from Sidwell Friends School (1982), and from Brown University (1986).
Question: When does Meet Me in the birthplace of Gracie's director take place?
Answer: starting with Summer 1903.
Answerable Example2:
Passage: Paragraph 1: The city has a Mayor and is one of the 16 cities and towns in England and Wales to have a ceremonial sheriff who acts as a
        deputy for the Mayor. The current and 793rd Mayor of Southampton is Linda Norris. Catherine McEwing is the current and 578th sherriff. The
        town crier from 2004 until his death in 2014 was John Melody, who acted as master of ceremonies in the city and who possessed a cry of 104
        decibels. Paragraph 2: John May (born 26 September 1849 in Southampton, Hampshire; date of death unknown) was an English cricketer. May was
        a right-handed batsman who was a right-arm fast bowler.
Question: Who is the current mayor of the birthplace of John May?
Answer: Linda Norris.
Un-answerable Example:
Passage: Paragraph 1: Imran Khan has held the office of Prime Minister since 18 August 2018, following the outcome of nationwide general
        elections held on 25 July 2018. Paragraph 2: Hampi, also referred to as the Group of Monuments at Hampi, is a UNESCO World Heritage Site
        located in east - central Karnataka, India. It became the centre of the Hindu Vijayanagara Empire capital in the 14th century. Chronicles
        left by Persian and European travellers, particularly the Portuguese, state Hampi was a prosperous, wealthy and grand city near the
        Tungabhadra River, with numerous temples, farms and trading markets. By 1500 CE, Hampi - Vijayanagara was the world's second - largest
        medieval - era city after Beijing, and probably India's richest at that time, attracting traders from Persia and Portugal. The
        Vijayanagara Empire was defeated by a coalition of Muslim sultanates; its capital was conquered, pillaged and destroyed by sultanate armies
        in 1565, after which Hampi remained in ruins. Paragraph 3: As of June 2018, the Government of Karnataka consists of 27 ministers including
        Chief Minister and a Deputy Chief Minister. Paragraph 4: Thekkady (Idukki district) is the location of the Periyar National Park, which is
        an important tourist attraction in the Kerala state of India.
Question: As of 2018, who is the minister of the state where hampi tourist place is located?
Answer: unanswerable.
```

Variant3

Figure 10: The three variants of in-context examples for the MuSiQue prompts. For the regular prompts, we use the two answerable examples, whereas for the prompts hinting at the possibility of (un)answerability, we use the first answerable example and the (un)answerable examples. Additionally, for the other prompt variants, we replace the "unanswerable" answer of the (un)answerable example with "IDK" and "N/A", accordingly.