# OpenReview forum: "The Curious Case of Hallucinatory (Un)answerability: Finding Truths in the Hidden States of Over-Confident Large Language Models"
_EMNLP/2023/Conference — EMNLP 2023 Main_

### Official Review · Reviewer_9CW2 · 2023-08-03

**Soundness:** 4

**Excitement:**

4: Strong: This paper deepens the understanding of some phenomenon or lowers the barriers to an existing research direction.

**Paper Topic And Main Contributions:**

This paper studies how/whether LLMs understand if they don't know the answer to a question given an input context (passage). They test three different approaches:

1. Prompt manipulation, where they provide a hint that the question could be unanswerable, and compare results with the case without the hint. They show that providing the hint significantly increases the results (up to 80%) in both zero-shot and few-shot settings.

2. Beam Relaxation, where they look for cues in the full beam indicating that the question is unanswerable (based on the existence of certain words in the decoded outputs). They show that increasing the beam size improves the task, when they consider a match in the full beam. However, the performance of the highest-ranking output doesn't increase by increasing the beam size.

3. Linear classification (using SVM) on the final hidden layer of the first generated token: Interestingly, they show that it is indeed possible to linearly separate the answerable and unanswerable questions. They do visualization using PCA which supports their claim. They also show that models learn this notation beyond a dataset (by training on one dataset and testing on others). They further show that if the linear separability is erased, the good results don't hold anymore.

These approaches perform very well on the three datasets they've used (Squad 2, Natural Questions and MuSiQue, where the 2nd and 3rd datasets are adapted for the task), where accuracy of unanswerable question detection is around around 75% to 92%, while the accuracy of the main task doesn't decrease because of applying these models. This indicates that LLMs do have the capability of detecting unanswerable questions.

**Questions For The Authors:**

See reasons to reject (2 and 3).

**Reasons To Accept:**

1. The paper studies an important problem and performs a thorough analysis with three approaches and on three datasets.

2. The linear separation method and its visualization is very interesting.

3. Their approaches could be used (probably with some improvements) to remove some hallucinations from the output.

4. Their processed datasets (if made available), and their setup could be used by others to improve the task.

**Reasons To Reject:**

1. While at the first glance, one might think the paper also considers open-domain settings, the setting is constrained to cases where the answer can only be found in the given context (but not in the already existing knowledge in the LLMs). I think this should be clarified.

2. The different approaches are not compared with each other. Also, it's not clear why few-shot settings do worse than zero-shot in many cases (see bold numbers in table 2b vs 2a).

3. I'm not sure if beam relaxation is done with the hint or without the hint?

4. The results of the first two approaches depend highly on the used keywords (Appendix C), but not much is said about it in the paper. Maybe a simple classifier using these initial keywords could improve the reliability of the results.

**Reproducibility:**

4: Could mostly reproduce the results, but there may be some variation because of sample variance or minor variations in their interpretation of the protocol or method.

**Reviewer Confidence:**

4: Quite sure. I tried to check the important points carefully. It's unlikely, though conceivable, that I missed something that should affect my ratings.

---

> ### Author Rebuttal · Authors · 2023-08-28
>
> We would like to thank the reviewer for the positive feedback and appreciation of our work.
> Here are our responses to the reviewer’s questions and comments:
>
> ***Closed-domain setting***
>
> We thank the reviewer for pointing out this issue, and would adapt the final version of the paper to clarify the fact that our work focuses solely on the closed-domain setting. We also agree that an open-domain setting could be an interesting follow up research, but it is outside the scope of our current work.
>
> ***Comparison between methods***
>
> We would like to note that our main goal was to demonstrate, via different approaches, LLM’s inherent understanding of unanswerability, even when they in fact hallucinate responses. However, following the reviewer’s comment, we would try to compare the approaches for the camera-ready.
>
> Regarding the reviewer’s concern about the few-shot setting performing worse than the zero-shot setting in many cases, we suspect that this is due to the fact that in-context-learning examples are selected at random and sometimes they are dissimilar from the target example enough to hinder performance. For the final version, we would try to address this phenomenon, by using several ICL examples and averaging over their results.
>
> ***Beam relaxation and hints***
>
> In our setting, the beam relaxation is done without the hint. We will clarify this in the final version of the paper.
>
> ***Unanswerability keywords***
>
> Indeed, the identification of answers indicating unanswerability depends on the keywords listed in Appendix C. As we mentioned in the footnote at the bottom of page 5, as well as in Appendix C, these keywords were collected based on an extensive analysis of the models' outputs (~500 of the unanswerable instances for each model). However, we agree with the reviewer's observation regarding the limited discussion in the paper of this analysis and will address this in the final version.

---

### Official Review · Reviewer_CavA · 2023-08-04

**Typos Grammar Style And Presentation Improvements:** line 224
**Soundness:** 4

**Excitement:**

4: Strong: This paper deepens the understanding of some phenomenon or lowers the barriers to an existing research direction.

**Paper Topic And Main Contributions:**

For question answering, this paper is to study the research problem of predicting question (un)answerability, given the pretrained language model, especially for LLMs.

Authors study this problem mainly from 3 perspectives, including (1) prompt manipulation, (2) beam relaxation and (3) embedding space probing, which would cover the major factors that influence the QA task in LLMs. Experiments are conducted properly with 3 LLMs (Flan-T5, Flan-UL2, and OPT-IML) on 3 benchmark datasets (Squad v2, NQ, and MuSiQue).

From the experiments, authors present their findings in each of the direction. (1) hint in prompt would much help to predict the unanswerability of the question; (2) larger beam size would help to recognize the unanswerable questions; (3) authors mainly analyze and visualize the embedding space of the first token for answer generation. The visualized analysis shows that the embedding space would be easier to be separated given hint prompt compared with regular prompt. Meanwhile, concept erasing tech is adopted to verify the influence of the answerability subspace.



**Questions For The Authors:**

In language generation, one problem is that the beam search based approach would make the top k generated text look identical. The reviewer wonder if authors also observe the same phenomena in beam relaxation?

While, in this paper, it looks that the (un)answerability prediction is much affected by the beam size. From Line 191, authors mention that the final layer of the first generated token is used for prediction. Does that mean the beam size will much change the first generated token? Appreciate if authors can elaborate more on this, and also please correct reviewer if the understanding is wrong.

**Reasons To Accept:**

(1) study the answerability problem from different perspectives

(2) hint prompt helps to predict the (un)answerability of the question

(3) visualized embedding space gives readers better illustration

Not necessarily negative points, more or less some discussion

Prompt engineering is important and useful. If pure prompt engineering is somewhat like black box testing. The LLMs are too big, which makes the white box testing too difficult. With the hint prompting, and analysis on the first generated token, is it somewhat like grey-box testing? Maybe this is a question for the community to answer together.

Please correct the reviewer if anything is missing or incorrect.

**Reasons To Reject:**




**Reproducibility:**

4: Could mostly reproduce the results, but there may be some variation because of sample variance or minor variations in their interpretation of the protocol or method.

**Reviewer Confidence:**

3: Pretty sure, but there's a chance I missed something. Although I have a good feel for this area in general, I did not carefully check the paper's details, e.g., the math, experimental design, or novelty.

---

> ### Author Rebuttal · Authors · 2023-08-28
>
> We would like to thank the reviewer for the positive feedback and appreciation of our work, as well as for spotting typos, which we will correct in the final version of the paper.
> Here are our responses to the reviewer’s questions and comments:
>
> ***gray-box testing***
>
> We agree with the reviewer that our analyses, especially those combining the prompt-engineering with the subspace exploration, can be seen as gray-box testing. We also hope that our work will lead to further such explorations in relation to unanswerability, to better understand and control its effects.
>
> ***Identical responses in beam-relaxation***
>
> Upon examining the final k candidates across instances, we did observe similar answers. Specifically, when a candidate indicated unanswerability (refer to Appendix C), we observed variations of unanswerability-indicating responses, while remaining candidates were similar. Alternatively, in cases when none of the candidates signified unanswerability, all the candidates resembled other candidates.
>
> ***Relation between the answerability prediction and beam-search***
>
> As the reviewer noted, for (un)answerability prediction, we made use of the last hidden layer of the first generated token. Since beam search impacts only post-first-token generation (because until the first generation, there hasn't been yet a separation into beams) and our predictions utilize embeddings that precede the generation of the first token, beam search does not influence our subspace exploration.

---

### Official Review · Reviewer_t5Sd · 2023-08-05

**Soundness:** 4

**Excitement:**

4: Strong: This paper deepens the understanding of some phenomenon or lowers the barriers to an existing research direction.

**Paper Topic And Main Contributions:**

This paper investigates LLMs' behaviours when presented with unanswerable questions, and explores different ways of improving LLMs in responding to these unanswerable questions. Authors explore four different approaches to either investigate LLM’s awareness toward “unanswerability” or to improve LLM’s ability to handle unanswerable questions: i) prompt manipulation by explicitly providing extra instruction “If it cannot be answered based on the passage, reply “unanswerable””. ii) “beam relaxation” which refers to looking for any cues of unanswerability in the beam candidates (i.e., keywords such as “unanswerable, IDK, I don’t know, N/A, none of the above, etc…). iii) training a linear classifier to see if answerability can be classified based on LLM representation. iv) erasing answerability subspace from LLM embedding space.

**Questions For The Authors:**

1. Why do we want to erase the answerability subspace..?
2. In Figure 3: although the trend persists across all datasets, I can observe relatively poor performance in both NQ and MuSiQue in comparison to the Squad dataset. What could be the reason for this?
3. In line 149, 150, you wrote: “three experimental approaches: i) prompt manipulation, 2) beam scrutiny, 3) probing”, but this does not seem to align with the following content (probing is missing). Is this a mistake?



**Reasons To Accept:**


1. **Important Problem:** Hallucination is an important and timely problem that needs to be solved for the safe deployment of LLMs. I especially value the fact that the authors are addressing hallucination from the unanswerability perspective.
2. **Effectiveness of proposed method:** Although prompt manipulation and beam relaxation methods are both simple, the performance improvement is drastic without harming the original performance.
3. **Comprehensive experiments:** Experiments are conducted on three different LLMs, and on three different datasets.
4. **Interesting insights:** This paper offers a detailed analysis of LLM from unanswerability that could be beneficial to the community.


**Reasons To Reject:**

There are no major concerns I have regarding this paper.
Here are some minor comments:

1. I do not really get what we can learn from “erasing the answerability subspace”.
2. Dividing Method section into subsections. Based on my understanding, prompt manipulation and beam relaxations go beyond just providing analysis of LLM behaviour towards unanswerability. They can be direct solutions to improve the problem of hallucination when presented with unanswerable questions. On the other hand, the other two subspace explorations are merely for understanding the LLM representation from unanswerability aspects. I believe separating them into two parts and emphasising the usefulness of prompt manipulation & beam relaxation method would strengthen this paper.


**Reproducibility:**

3: Could reproduce the results with some difficulty. The settings of parameters are underspecified or subjectively determined; the training/evaluation data are not widely available.

**Reviewer Confidence:**

3: Pretty sure, but there's a chance I missed something. Although I have a good feel for this area in general, I did not carefully check the paper's details, e.g., the math, experimental design, or novelty.

**Typos Grammar Style And Presentation Improvements:**

- There are grammar/typo issues in this paper. I recommend doing another pass through the paper to ensure these issues are resolved. For instance, “tried to use utilise beam search…” (line 131), “affective methods” (line 224) —> effective methods.

---

> ### Author Rebuttal · Authors · 2023-08-28
>
> We would like to thank the reviewer for the positive feedback and appreciation of our work.
> Here are our responses to the reviewer’s questions and comments:
>
> ***Why are the erasure experiments needed?***
>
> The probing experiments alone show that *there exists* a subspace in the representation space that encodes answerability. This, however, does not necessarily show that the subspace is being utilized by the model when deciding whether or not to output an answer. In other words, the information can be present in the representation and at the same time be irrelevant to the model’s behavior [1]. By erasing this subspace and examining the effect on the model’s output, we can verify that the subspace we identify is behaviorally meaningful, opening an avenue for performing interventions in that subspace in order to modify the model’s behavior.
>
> [1] Elazar, Yanai, Shauli Ravfogel, Alon Jacovi, and Yoav Goldberg. "Amnesic probing: Behavioral explanation with amnesic counterfactuals." Transactions of the Association for Computational Linguistics 9 (2021): 160-175.
>
> ***Dividing Method section into subsections***
>
> We agree with the reviewer’s comment that such a division might be useful. However, as highlighted in lines 491-494, an answerability classifier *can*, in fact, be used to filter unanswerable questions, in a reliable manner which also transfers between datasets (Figure 7). As such, we do not see it as a mere analysis tool. Nevertheless, in light of the reviewer’s comments, we would consider a better layout of the sections for the final version of the paper.
>
>
> ***Figure 3  -relatively poor performance in both NQ and MuSiQue compared to Squad:***
>
> We hypothesize that the trends towards decreased performance are due to distribution shift and a more challenging setting:
> * NQ’s queries do not always appear in a question format. LLMs trained on QA datasets, were mostly introduced to unanswerable questions via datasets like Squad - where questions are in a question format. This distribution shift may cause decreased performance.
> * MusiQue dataset’s requirement for multi-hop poses a significant challenge - theren’t many such datasets on which models could have been trained, and even less so with unanswerable questions. This is evident also in tables 2&3.
> Furthermore, this decreased performance can be observed in the model’s decreased performance even over answerable questions (tables 6&8), which further underscores the challenging nature of these datasets, compared with Squad.
>
> ***three experimental approaches vs. 4 in the paper:***
>
> By probing we refer to sections 3.3 and 3.4. We will clarify this towards the final paper version.

---

### Meta-Review · Area_Chair_9yLy · 2023-09-13

**Recommendation:** 4

**Metareview:**

**Strengths**:

1. Paper studies a critical problem of detecting hallucination in LLMs from an unanswerability perspective.

2. 3 methods are proposed: (i) prompt engineering to check for unanswerability (ii) beam relaxation: checks for unanswerability cues in the generated candidates using cue phrase list. (iii) linear SVM classifier on first generated token to check for answerability.

3. Experiments with three different LLMs, and three different datasets.

**Weaknesses**:

1. Proposed methods are very simple.

2. Not clear to me even from the rebuttal as to how erasing experiments can help in performing interventions in that subspace in order to modify the model’s behavior.

3. There is not enough analysis: The different approaches are not compared with each other. Also, it's not clear why few-shot settings do worse than zero-shot in many cases

**Suggestions**:

1. Please handle typos. There are too many.

2. Explain motivation for erasing experiments more clearly.

3. "relatively poor performance in both NQ and MuSiQue compared to Squad" -- please include your explanation in the main paper.

4. Limitations section should include: (a) expts on QA with context only. (b) lack of comparison across methods. (c) Small cue phrase list (d) 3 datasets and 3 models. Need more expts to generalize to other datasets/models.

---

### Decision · Program_Chairs · 2023-10-07

**Decision:**

Accept-Main

**Comment:**

**Strengths**:

1. Paper studies a critical problem of detecting hallucination in LLMs from an unanswerability perspective.

2. 3 methods are proposed: (i) prompt engineering to check for unanswerability (ii) beam relaxation: checks for unanswerability cues in the generated candidates using cue phrase list. (iii) linear SVM classifier on first generated token to check for answerability.

3. Experiments with three different LLMs, and three different datasets.

**Weaknesses**:

1. Proposed methods are very simple.

2. Not clear to me even from the rebuttal as to how erasing experiments can help in performing interventions in that subspace in order to modify the model’s behavior.

3. There is not enough analysis: The different approaches are not compared with each other. Also, it's not clear why few-shot settings do worse than zero-shot in many cases

**Suggestions**:

1. Please handle typos. There are too many.

2. Explain motivation for erasing experiments more clearly.

3. "relatively poor performance in both NQ and MuSiQue compared to Squad" -- please include your explanation in the main paper.

4. Limitations section should include: (a) expts on QA with context only. (b) lack of comparison across methods. (c) Small cue phrase list (d) 3 datasets and 3 models. Need more expts to generalize to other datasets/models.